# Transferring Task Goals via Hierarchical Reinforcement Learning

**Saining Xie[1,2], Alexandre Galashov[2], Siqi Liu[2], Shaobo Hou[2],**
**Razvan Pascanu[2], Nicolas Heess[2], Yee Whye Teh[2]**
[1]UC San Diego, La Jolla, USA, [2]DeepMind, London, UK

## 1 Introduction

An important property of our way of interacting with the real world is that we can, in many cases, form a representation of *what* to do separately from *how* to do it. And when our preferred way of solving a task is no longer feasible, we can often retrain ourselves to solve a task differently, using our high-level understanding of what needs to be done. In this paper we study the potential benefits of a separation of concerns between high-level task understanding and low-level actuation in the context of a hierarchical reinforcement learning system consisting of a high-level controller that sets subgoals associated with a guiding reward, and a low-level controller that achieves these subgoals. The low-level controller is trained to maximize the guiding reward, while the high-level is trained to maximize task reward. Both parts of the system can be trained simultaneously or separately.

Endowing an agent with an ability to separate task understanding from task execution can be advantageous for a variety of reasons, and there has been significant recent interest (Heess et al., 2016; Frans et al., 2017; Hausman et al., 2018; Florensa et al., 2017; Denil et al., 2017; Andreas et al., 2016) e.g. under the banner of hierarchical RL or motor primitives in learning re-usable low-level policies, in order to learn new tasks faster, or to learn multiple tasks with the same body (See Appendix B for a summary). In this paper we focus on one specific perspective, whereby the high-level controller in a HRL system is re-used, while the low-level controller is retrained, in order that the same task can be solved with different bodies or if the body changes. There is a number of important scenarios for this perspective, and some of them have been discussed in recent works (Gupta et al., 2017; Devin et al., 2016). Firstly, for an embodied agent it is natural to expect that the body will undergo changes e.g. due to damage or wear-and-tear. Secondly, manufacturing imperfections, e.g. with cheaper and thus lower quality hardware, often mean that different instances of the same body can exhibit variations significant enough to affect performance when using a single learned controller. Thirdly, the controller may be trained in simulation before being applied to physical bodies (Rusu et al., 2016; Peng et al., 2017a). While the high-level goals are the same in simulation and on real hardware, there can be significant changes in the low-level dynamics and perceptual features.

In all these scenarios, the low-level controller will have to adapt while the high-level subgoals remain unchanged. This low-level adaption should be achieved faster than retraining the whole controller. Intuitively, the high-level subgoals should provide more direct learning signals that can be quickly used by the low-level controller with less reliance on rewards from the environment. To illustrate this general idea we consider a simplified scenario in which we assume that an invariant observation space (for the high-level controller) and a plausible set of subgoals are available (See Appendix C).

We consider multiple locomoting bodies in a navigation environment and show that: 1) the system can indeed be trained end-to-end to solve non-trivial tasks, 2) the subgoals with associated guiding rewards can lead to faster learning than relevant baselines since this setup simplifies the credit-assignment problem, 3) the resulting system achieves robustness to changes in the low-level actuation including damage since the system can be rapidly re-trained using only guiding rewards (i.e. no task reward), 4) the high-level controller can be transferred across very different body types.

**Architecture.** We design our control architecture so as to achieve a separation between the high-level task objectives and the low level control, i.e. between the *what* and the *how*. The model consists of a two layer control hierarchy comprising a **high-level (HL) controller** and a **low-level (LL) controller**. The HL controller specifies subgoals $g \in \mathcal{G}$ to the LL controller. The LL controller produces body-specific control signals in response. In each environment state $s$, an agent receives observations $o$ of various types. We assume that we can divide these observations into two components. The first component $o^L$ consists of the "body-centric" observations. For a legged robot this includes, for instance, joint angles, joint velocities, and various other low-level on-board sensors such as

accelerometer, velocimeter, or gyroscopes. It also includes ray-casting information, which provide distances to objects from the body. The second component $o^H$ consists of observations that provide global, body-independent but task-relevant information, e.g. top-mounted cameras or minimaps.

To achieve the desired separation of concerns and invariance to details of the body and actuation scheme we use two strategies: Firstly, we shield the HL controller from LL controls, by not providing it with the body-specific information. Secondly, we provide the LL controller with body-centric information, but do not train it directly to maximize task reward. Instead we associate each subgoal with a guiding reward which the LL controller aims to maximize. In our scheme only the HL controller is trained to maximize overall task reward.

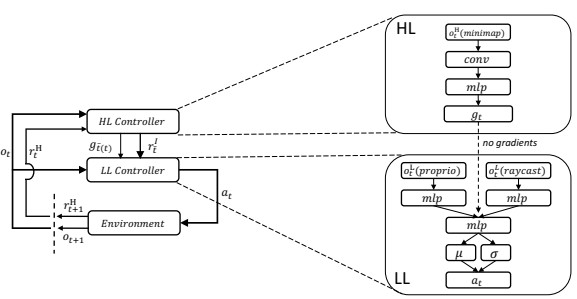

Figure 1: Schematic of our architecture.

**Environment.** All our experiments are conducted in a physical simulation environment implemented in Mujoco (Todorov et al., 2012). We used a more difficult version of the *walls* course similar to (Heess et al., 2017) (Figure 2 (right)) as the testbed for our HRL system. The difficulty comes from the longer wall obstacles that agents have to navigate around. To show that our HRL system is able to transfer high-level task understanding across bodies with different properties and capabilities, we consider two torque-controlled walkers: *Ball*, an easy-to-control body (Figure 2 (left)), and *Ant*, a harder-to-control quadrupedal body (Figure 2 (middle)). These two body types represent two degrees of difficulty for learning the LL controller. We also designed two "damaged" bodies: *Damaged Ball*, where we invert the gear of the *Ball* joints (as a result, control orders are also inverted), and *Damaged Ant*, where we freeze some leg joints of the *Ant* walker. As a result, the damaged Ant will need to adapt to walking using the undamaged legs.

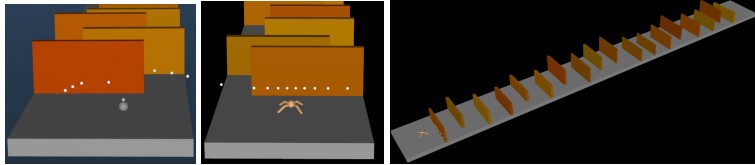

Figure 2: Visualization of different bodies (an easy-to-control Ball and a hard-to-control Ant), and the *walls* course for the navigation task. White dots show the egocentric rays cast by the walker.

## 2  RESULTS

**HRL vs. Baselines** We consider several *Baseline* models: (1) a two-stream network similar to the LL controller in our HRL architecture (Figure 1), but trained with the task reward; (2) a similar two-stream network trained with task rewards but that also receives the minimap (See Appendix C) as observations; and (3) the HRL setup but without guiding rewards. For the *Ball*, a body that is simple to control, we only observe a small advantage of the hierarchical setup (Figure 3(a)) over the

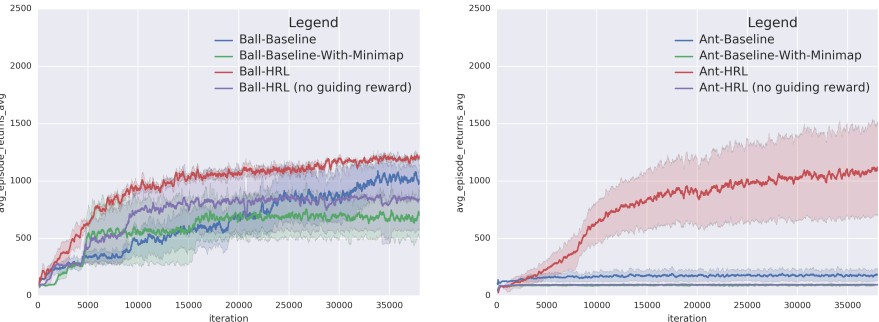

Figure 3: **HRL benchmark performance** on *Ball* (left) and *Ant* (right).

baselines. For the *Ant* (Figure 3(b)), we see a more pronounced advantage of using the HRL system. The performance relative to the additional baselines (2) and (3) suggests that the advantage is indeed due to the guiding rewards rather than the difference in observation spaces.

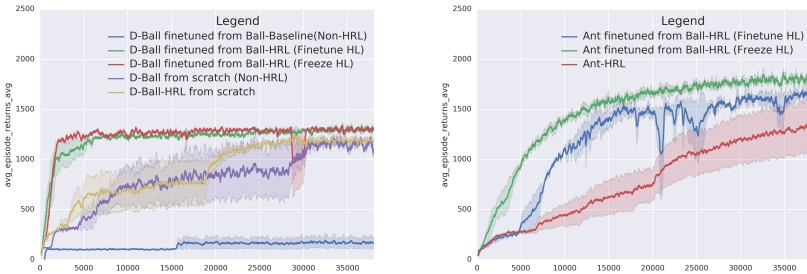

Figure 4: **Left**: *Ball* to *Damaged-Ball* (D-Ball) transfer. **Right**: *Ball* to *Ant* transfer.

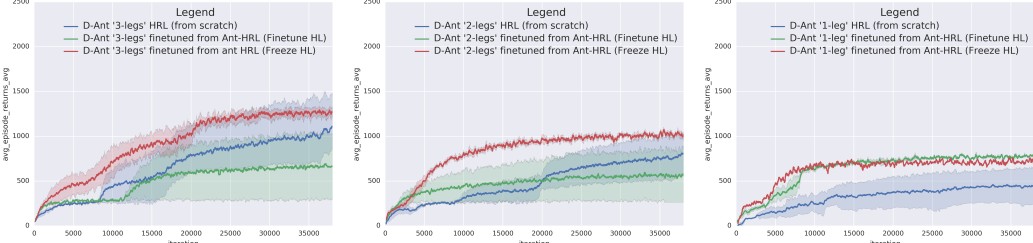

Figure 5: *Ant* to *damaged Ant* (D-Ant) transfer. **Left to Right**: Damaged quadrupedal *Ants* navigate with three, two, and one leg(s) available.

**Transferring Task Goals to Other Bodies** In this section we present the results for two transfer scenarios: (1) transferring from a fully functional body to a damaged one (*Ball/Ant* to *damaged Ball/Ant*) and (2) transferring between bodies of different complexities (*Ball* to *Ant*).

We show *Ball* to *damaged Ball* transfer results in Figure 4 (left). In this setting, compared to the training from scratch baseline, transferring the HL task goals leads to significantly faster learning on a new (damaged) body. For a non-hierarchical *Baseline* model, finetuning after pre-training on a normal body is actually slower than learning from scratch, since the agent needs to unlearn its learned behavior first before adapting to a new body. We consider transferring from a fully functional *Ant* body to a damaged one too. The *damaged Ant* has one, two or three (out of four) leg joints frozen. These modifications increase the difficulty of the task considerably. Transferring the task goals, by means of transferring the HL controller trained on a normal body, allows the damaged Ants to quickly adapt to their body changes, and successfully learn to solve the navigation task, in the extreme scenario with only one usable leg. As for the *Ball* we observe faster learning, compared to learning controllers for the damaged Ants from scratch. Finally, we investigate the possibility of transferring the HL controller trained on the *Ball* to the *Ant* body. This can be seen as a curriculum over body complexity: we first learn to solve the task with a simpler body before transferring the HL controller to a more complex one. From the results in Figure 4 (right), we observe that by transferring task goals with a frozen HL controller, learned on a simple *Ball* body, our method outperforms training from scratch. This is an encouraging result in that it suggests that in certain situations a complex task can be solved first with an easy-to-control body and that the understanding gained in the process can then be used to facilitate learning with a more complex body.

We observe that freezing the HL controller (in contrast to finetuning it) is in general the better performing choice. Intuitively, on a new body, the initial noisy behavior of the LL controller makes the gradient updates to the HL controller unstable, potentially ruining the learned HL policy. However, for simple bodies/systems the LL response to the subgoals is easy to learn and the high-level controller's performance therefore does not degrade.

**Conclusions.** Our results suggest that there can be a benefit to introducing a separation of concerns in the form of subgoal specification vs. subgoal execution between a high-level and a low-level controller, allowing the transfer high-level task understanding across bodies of different morphologies. To demonstrate these benefits we made two simplifying assumptions: the availability of a suitable subgoal space and of a body-invariant perceptual space. Relaxing these assumptions will be the topic of future work.

ACKNOWLEDGMENTS

We would like to thank Josh Merel, Yuval Tassa, Greg Wayne, Jay Lemmon, Pushmeet Kohli, Raia Hadsell and Koray Kavukcuoglu for many insightful and influential conversations.

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

## A    HRL FORMULATIONS

Here we detail the formulations for the HRL system used in our work.

To encourage the HL controller to learn over longer time scales, we set it up to sample a new subgoal $g_t$ for the LL controller only once every $k$ LL controller steps. In other words, for $t = 0, k, 2k, \ldots$ we draw $g_t$ from the HL policy $\pi^H(\cdot|o^H_{\leq t})$ given the history of past global observations $o^H_{\leq t}$. The LL controller operates at every time step and samples body-specific actions $a_t \sim \pi^L(\cdot|o^L_{\leq t}, g_{\bar{t}(t)})$ from the LL policy, where $o^L_{\leq t}$ are the body-centric observations up to time $t$ and $g_{\bar{t}(t)}$ is the most recently sampled subgoal. Together, the HL and LL controllers define the policy $\pi$ for the agent.

The HL controller maximizes discounted task return

$$\mathcal{J}^H = \mathbb{E}_{\pi(\tau)} \left[ \sum_{t \in \{0, k, 2k, \ldots\}} (\gamma^H)^{t/k} \left( \sum_{t'=t}^{t+k-1} r^H(s_t, a_t) \right) \right]$$

where $r^H(s_t, a_t)$ is the task rewards and $\tau = (s_0, g_0, a_0, s_1 \ldots, s_k, g_k, a_k, \ldots)$ is the trajectory sampled jointly from $\pi^H$ and $\pi^L$ following the procedure described in the preceding paragraph. Since the subgoal is updated only every $k$ steps, we choose to discount the rewards only every $k$ steps.

The LL controller is trained to maximize the guiding rewards $r^L(o^L_{\leq t}, a_t, g_{\bar{t}(t)})$:

$$\mathcal{J}^L = \mathbb{E}_{\pi(\tau)} \left[ \sum_{t=0}^{T} \left[ \prod_{t' \leq t} \gamma^L_{t'} \right] r^L(o^L_{\leq t}, a_t, g_{\bar{t}(t)}) \right]$$

Note that the guiding reward is a function of the most recent subgoal specified by the HL controller, and can depend on the observations in which that subgoal was chosen. Furthermore, we allow for a (possibly time-varying) discount factor that can deviate from the discount factor associated with the task.

The above scheme can be implemented in different ways. In particular, the subgoal space $\mathcal{G}$ can be either continuous or discrete. In most of our experiments the subgoals are associated with movement in discretized directions, such as "moving forward", "moving left" or "moving right" (see Section C). Furthermore, the low-level learning problem can be seen as a sequence of independent tasks imposed by the high-level controller, or as an ongoing parameterized task.

Further points are noteworthy about this setup: Firstly, due to the particular choice of discounting of the task reward the effective horizon of the learning problem for the high-level controller can be very long. Secondly, the learning problem for the low-level controller is effectively task independent. Thirdly, in this setup the horizon for the high-level and the low-level learning problems can be decoupled. In particular, credit assignment to the low level can be nearly immediate while the high-level can still take long-term task reward into account. Finally, for training of the low-level controller the task reward is not needed. The model can be optimized using policy gradients for both levels. From the perspective of the HL controller, the LL behavior is given and learning $\pi^H$ reduces to policy gradient with action space $\mathcal{G}$:

$$\nabla_\theta \mathcal{J}^H = \mathbb{E}_{\pi(\tau)} \left[ \sum_{t \in \{0, k, 2k, \dots\}} \nabla_\theta \log \pi^H(g_t | o_{\leq t}^H)(R_t^H - b^H(o_{\leq t}^H)) \right]$$

where $R_t^H = \sum_{t' = \{t, t+k, t+2k, \dots\}} (\gamma^H)^{(t'-t)/k} \bar{r}_{t'}$ and we use the shorthand $\bar{r}_t = \sum_{t'=t}^{t+k-1} r^H(s_{t'}, a_{t'})$. $b^H$ is the baseline. For the LL controller we similarly have

$$\nabla_\psi \mathcal{J}^L = \mathbb{E}_{\pi(\tau)} \left[ \sum_t \nabla_\psi \log \pi^L(a_t | o_{\leq t}^L, g_{\bar{t}(t)})(R_t^L - b^L(o_{\leq t}^L, g_{\bar{t}(t)})) \right]$$

where the low level return is

$$R_t^L = \sum_{t' \geq t} \left[ \prod_{t''=t+1}^{t'} \gamma_{t''}^L \right] r^L(o_{\leq t'}^L, a_{t'}, g_{\bar{t}(t')})$$

and $b^L$ is the current subgoal dependent baseline. Note in both cases the expectations are taken with respect to both high and low-level actions. If either controller is being optimized then the other controller can be seen as forming part of the (stochastic) environment.

## B  RELATED WORK

There has been a large amount of interest in hierarchical or otherwise structured network architectures that allow reusing previously acquired behaviors across tasks and/or for learning faster in the first instance.

There is a large body of classical work on hierarchical approaches in reinforcement learning (Dayan & Hinton, 1993; Sutton et al., 1999; Precup, 2000; Dietterich, 2000; Sutton, 1995; Boutilier et al., 1997; Dayan, 1993; Parr & Russell, 1998; Precup et al.; 1998; Wiering & Schmidhuber, 1997).

Recent works such as (Heess et al., 2016; Frans et al., 2017; Hausman et al., 2018; Florensa et al., 2017) have focused on learning and transferring reusable low-level skills using respectively information hiding, meta-learning, and information-theoretic regularization to induce a separation of concerns in the architecture.

(Vezhnevets et al., 2017) has been an inspiration for our work. They propose an architecture in which a high-level controller explicitly sets subgoals for and provides appropriate rewards to a low-level controller. However, in their work observations are unstructured and goals are discovered in an arbitrary feature space. The resulting high-level controller is unlikely to be suitable for cross body transfer without additional regularization.

(Devin et al., 2016) propose an architecture that enforces a separation between task-specific and body-specific modules which can then be combined in novel ways. (Denil et al., 2017) employ a similar idea but focus on achieving new task variations rather than novel task-body combinations. (Gupta et al., 2017) induce an invariant feature space in which skills for different bodies can be

represented. (Devin et al., 2016; Heess et al., 2016; Gupta et al., 2017) make similar assumptions to our work regarding a separation of body-specific and body-independent observations.

While not directly concerned with learning transferable subgoals, (Andreas et al., 2016) is related in spirit in that they fix the high-level structure associated with tasks (i.e. their decomposition into sub-policies) to induce reusable policy components.

There has also been a series of works learning controllers that are robust to changes in the dynamics or allow for limited structural modifications to the body. For instance wang2018nervenet have trained graph-structured controllers mimicking the topology of the body that exhibit a certain degree of robustness to changes in the structure of the body. (Yu et al., 2017; Rajeswaran et al., 2016; Peng et al., 2017b) trained controllers to exhibit some robustness to changes in the underlying dynamics of the model by exposing them to many different conditions. (Yu et al., 2017) combined this idea with explicit online system identification.

## C  EXPERIMENTAL SETUP DETAILS

**Task reward.** While carefully designed reward shaping can help improving learning, we rely on the a simple environment reward defined in Heess et al. (2017), which is proportional to the instantaneous velocity along the x-axis.

**Guiding reward and subgoals.** The HL subgoals $g_t$ are chosen from the eight (inter-)cardinal directions illustrated in Figure 6(right). Given a subgoal $g_t$ drawn at time $t$, the guiding reward for the LL controller to move in direction $g_t$ is defined as:

$$r_t^I = -\frac{1}{k}\sum_{i=0}^{k-1} dist(\bar{o}_{t+i} - \bar{o}_t, g_t), \tag{1}$$

where $dist$ is a quasi-metric, $\bar{o}_t$ represents the agent's global coordinates $(x, y)$, which can be extracted from global observations $o^H$. For $dist$ we use

$$dist(\bar{o}_{t+i} - \bar{o}_t, g_t) = -(\bar{o}_{t+i} - \bar{o}_t)^\top \frac{g_t}{\|g_t\|},$$

but experimented with different functions (See Appendix D). These subgoals and guiding rewards ground the communication channel between HL and LL controllers and encourage the LL controller to move in the direction given by $g_t$.

**Observations for LL.** The LL controller makes use of body-centric observations from the environment. The egocentric ray-casting is visualized in Figure 2. The walker projects $N = 60$ rays from ray-casters attached to its body, in directions spread out over a pre-defined field of view (120 degrees). Ray-casting returns the distance of the object hit by each ray.

**Observations for HL**. In this work we directly construct global observations that are body-independent, for which we designed a "minimap" observation Figure 6(left). At each time step, the minimap represents a top-down snapshot of the navigation environment. Since the course is long, the minimap is a moving window along the x-axis, covering about a third of the entire course. The minimap is a essentially sparse binary matrix that indicates the positions of the walls as well as the walker's position. It is worth noting that the use of the minimap is purely for the purpose of constructing a body-invariant input for the HL controller. We have shown empirically in Figure 3 that using minimap alone does not give a performance edge over using terrain feature map observations. The HL and LL controllers also receive global body orientation and position coordinates.

## D  ADDITIONAL STUDIES

**Learning horizon for HL** We investigate the influence of the time horizon ($k$ in Equation (1)). Figure 7 shows that values of $k$ larger than 1 has some advantage for complex bodies like *Ant*, indicating that learning with longer-term horizon in HL controller has some benefit. For the simpler *Ball* body, the difference is insignificant.

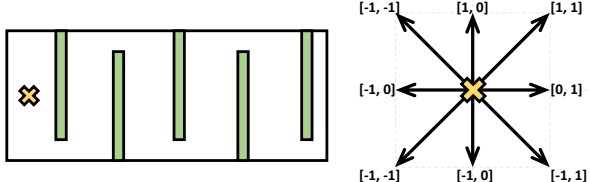

Figure 6: **HL observation and subgoals. Left**: a top-down "minimap" is constructed as the invariant HL input. **Right**: grounded subgoals (discrete and normalized directions). Domain knowledge is used as we assume the walker navigates itself on a 2-D plane.

**Quasi-metric for the guiding reward** We also investigate different quasi-metric functions for the *dist* function in the guiding reward. We test *cosine*, *projection* and *Euclidean* distances, which are defined as:

$$dist(\mathbf{a}, \mathbf{b}) = \begin{cases} -\frac{\mathbf{a} \cdot \mathbf{b}}{|\mathbf{a}||\mathbf{b}|} & \text{(cosine)} \\ -\mathbf{a} \cdot \frac{\mathbf{b}}{|\mathbf{b}|} & \text{(projection)} \\ ||\mathbf{a} - \mathbf{b}|| & \text{(Euclidean)} \end{cases} \tag{2}$$

Empirical results in Figure 7 suggest that the *projection* distance, given by projecting the movement vector $\bar{o}_{t+i} - \bar{o}_t$ onto the subgoal vector $g_t$ works best. This distance encourages the LL controller to fulfill the HL subgoals, by not only rotating to the correct direction, but also making progress along that direction; Cosine distance does not have such property as it loses the magnitude of the movement vector. The Euclidean distance sets absolute goals where the walker might not be able to reach, and it is not robust to changes in velocity scale.

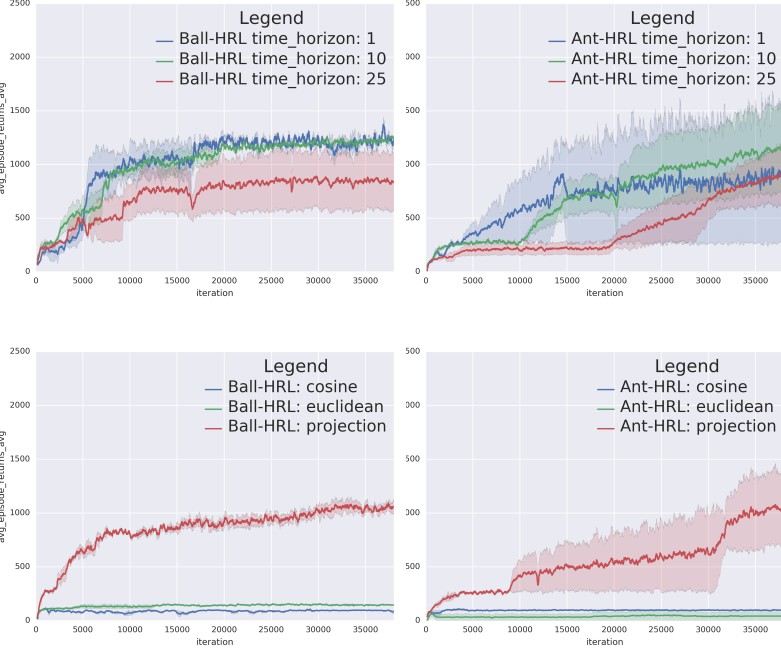

Figure 7: **Additional studies. Top Row**: different HL time horizons. **Bottom Row**: different quasi-metrics for guiding reward.

## E  VIDEO DEMONSTRATIONS

We show video recordings to demonstrate the behavior of the trained agents in our experiments:

- (1) Normal Ant body trained with our HRL system (`https://www.youtube.com/watch?v=PxuIiNzNCzI`)
- (2) Damaged Ant navigating with 3 Legs, with task goals transferred from (1) (`https://www.youtube.com/watch?v=RDINb5GwgKA`)
- (3) Damaged Ant navigating with 2 Legs, with task goals transferred from (1) (`https://www.youtube.com/watch?v=3UZOzWYv7e8`)
- (4) Damaged Ant navigating with 1 Leg, with task goals transferred from (1) (`https://www.youtube.com/watch?v=rFs3iOQtLi4`)
- (5) Normal Ball body trained with our HRL system (`https://www.youtube.com/watch?v=12YUeuhGTjM`)
- (6) Damaged Ball navigating with inverted gears, with task goals transferred from (5). (`https://www.youtube.com/watch?v=3O19wIJvsFc`)

