# OpenReview forum: "Transferring Task Goals via Hierarchical Reinforcement Learning"
_ICLR.cc/2018/Workshop — Reject_

### Official Review · AnonReviewer3 · 2018-03-08
**of moderate interest**

**Rating:** 5
**Confidence:** 4

**Review:**

The paper evaluates the benefit of hierarchical reinforcement learning,
in the context of being able to reuse a high-level controller, while being able
to introduce and learn new low-level controllers.  This is evaluated on a 4-legged ant
(and transferred to its damaged variations) and a steered ball (and transfer to the ball with
reversed steering).  The LL goals (output by the HL controller) consist of a specification of
one of 8 discrete steering directions.

Transfer learning is important and hierarchical structures are a logical (and common)
approach to apply in support of transfer learning.
The technical aspects of the paper, i.e., the experimental details, are well written.
The challenge for this paper is with positioning it with respect to the relative
abundance of related work in this area, and elucidating the differences and benefits
of the proposed approach. The specification of a "direction to steer in" is in many ways
an obvious choice of subgoal representation.

Many tasks may not have a clean separation between the high-level and low-level goals,
i.e., it will the case that a representation that allows for transfer will also be one that
has to make significant compromises.  Also, they may not always separate nicely into
"what" and "how" concerns. It would have been interesting to see how well steering direction
would work given agents that have significantly different speed capabilities, i.e., the
target direction may be the right command for a slow agent, but it may be unwieldy for an
agent which can build significant momentum, namely travel faster, but therefore also turn more
slowly.

[Andreas et al 2016]  (and other similar references) could be cited in their final
publication format, i.e., Proc ICML 2017 for this paper.

[Xue Bin Peng 2017a] and [Xue Bin Peng 2017b] are duplicates.
Is one of these intended to be DeepLoco, also by Xue Bin Peng?
That seems relevant for the work in this submission, given
a similar setup, i.e., specified sub-goal structure, and an ability to reuse HL
controllers across multiple LL controllers.

It would be interesting to comment on the state distributions encountered by both the LL
and HL controllers. Ideal transfer would occur if they don't influence each other much.
And where it could begin to fall apart is where they are found to be interdependent in a
strong way, implying that changing LL would point to a need to retrain HL, and then
needing to retrain/fine-tune the LL, and so forth.

It would be exciting if the HL could learn how to best reward the LL, i.e., to learn the sub-task representation.

Summary:
+ transfer learning problem
+ writing
- articulation of difference/benefit wrt prior art
- results are specific to "steering direction" (SD) subgoals, and relatively "easy" tasks, i.e., likely to be benefit from the SD subgoals and SD rewards.

---

### Official Review · AnonReviewer2 · 2018-03-09
**Simple approach works on toy problem, but general insights questionable**

**Rating:** 5
**Confidence:** 3

**Review:**

Pros:
+ Approach is fairly simple to try
+ Experiments show significant improvement in rewards over baseline

Cons:
+ The high-level controller seems very simple and learning it jointly with RL may not be that interesting
+ The idea may be a bit too general or vague
+ Novelty on the low side

The submission proposes solving a reinforcement learning task consisting of a high-dimensional robot navigation task using a deep policy gradient method that decomposes the system into high-level and low-level controllers.  These controllers are trained simultaneously, but with different reward functions.  The high-level controller produces subgoals for the low-level controller, and an auxiliary reward function rewards the low-level controller for achieving these subgoals.  Experiments also show that the learned high-level controller is robust to changes in the low-level model.

On the positive side, this is a relatively simple idea that leads to significant improvements in performance in the problem domain, and the goal of transferring the learned high-level controller to different bodies seems to have been achieved.

On the other hand, and from a more philosophical perspective, I question how valuable these kinds of results are in practice.  Showing that an RL agent can learn a policy from low-level observations to low-level controls from scratch is certainly interesting in a way, but introducing more structure such as this high-level / low-level divide seems to erode the novelty of that somewhat.  In particular, is it really necessary to learn the high-level controller simultaneously with the low-level controller?  The high-level controller is basically just learning to navigate a 2D maze, and in practice, isn’t it a bit silly to not use A* for this purpose? Granted, there is also the perception component, but as long as we are doing some mostly-decoupled form of learning, why not use some more direct form of navigation-learning, like inverse reinforcement learning with A* for inference?  Yes, there are caveats to this too, but it just seems like the set of situations in which this approach would be interesting is rather narrow.

I think if the application were something more interesting and realistic, then that would be another matter, since there could be significant novelty in the way the high-and-low-level controllers might be engineered.  As it stands, however, it is unclear how much insight this proposal would provide for such a problem.

---

### Official Review · AnonReviewer1 · 2018-03-12
**Not particularly novel, but a useful and thorough experimental analysis**

**Rating:** 7
**Confidence:** 4

**Review:**

The paper presents a method for transferring skills between different agent platforms (either morphology or actuation capabilities) by training a body-agnostic high-level controller and a goal-agnostic low-level controller.

The problem tackled is well-motivated and the evaluation is thorough for a workshop track submission. The idea of high and low level controllers has extensively been explored before (as the paper points out) and it is not clear this paper offers a significantly novel contribution to this area of literature. One of the reasons the approach works is because it employs a human-designed subgoal space and a body-invariant action and observation space. The first assumption to me greatly limits the generality of the method. However, the experimental analysis performed could be useful to the community working in the area of hierarchical reinforcement learning and the paper worthy of publication in the workshop track.

---

### Decision · Program_Chairs · 2018-03-20
**ICLR 2018 Workshop Acceptance Decision**

**Decision:**

Reject

**Comment:**

Based on the reviews, this paper has not been accepted for presentation at the ICLR workshop. However, the conversation and updates can continue to appear here on OpenReview.